# Blenderised Tube Feeds vs. Commercial Formula: Which Is Better for Gastrostomy-Fed Children?

**DOI:** 10.3390/nu14153139

**Published:** 2022-07-29

**Authors:** Neha Chandrasekar, Kate Dehlsen, Steven T. Leach, Usha Krishnan

**Affiliations:** 1School of Women’s and Children’s Health, University of New South Wales, Sydney, NSW 2052, Australia; s.leach@unsw.edu.au; 2Department of Nutrition and Dietetics, Sydney Children’s Hospital, Randwick, NSW 2031, Australia; kate.dehlsen@health.nsw.gov.au; 3Department of Paediatric Gastroenterology, Sydney Children’s Hospital, Randwick, NSW 2031, Australia

**Keywords:** blended tube feeding, commercial formula, enteral feeding, gastrostomy, gut microbiome, paediatrics

## Abstract

Blenderised tube feeds (BTF) have become a popular alternative to commercial formula (CF) for enterally fed children. This study sought to compare gastrointestinal (GI) symptoms, GI inflammation, and stool microbiome composition between children receiving BTF or CF. This prospective cohort study involved 41 gastrostomy-fed children, aged 2–18 years, receiving either BTF (*n* = 21) or CF (*n* = 20). The Paediatric Quality of Life Inventory Gastrointestinal Symptoms Scale (GI-PedsQL) was used to compare GI symptoms between the groups. Anthropometric data, nutritional intake, nutritional blood markers, faecal calprotectin levels, stool microbiota, and parental satisfaction with feeding regimen were also assessed. Caregivers of children on BTF reported greater GI-PedsQL scores indicating significantly fewer GI symptoms (74.7 vs. 50.125, *p* = 0.004). Faecal calprotectin levels were significantly lower for children receiving BTF compared to children on CF (33.3 mg/kg vs. 72.3 mg/kg, *p* = 0.043) and the BTF group had healthier, more diverse gut microbiota. Subgroup analysis found that 25% of caloric intake from BTF was sufficient to improve GI symptoms. The CF group had better body mass index (BMI) z-scores (−0.7 vs. 0.5, *p* = 0.040). Although growth was poorer in children receiving only BTF in comparison to the CF group, this was not seen in children receiving partial BTF. A combination of BTF and CF use may minimise symptoms of tube feeding whilst supporting growth.

## 1. Introduction

Children with chronic diseases limiting their ability to feed orally require supplemental nutrition via a gastrostomy [1]. Commercially produced nutritional formulae (CF) are generally delivered to children requiring enteral nutrition (EN). Formulas are commonly used as they have precisely quantified nutrients, are sterile, and are easy to administer for caregivers [2]. However, administering CF via gastrostomy may induce adverse gastrointestinal symptoms [3,4] and has been implicated in reducing the diversity of microbial species in the gut microbiome [5]. Additionally, CF is characteristically rich in saturated fats and added sugar, lacks fibre, and contains preservatives required for shelf stability [6]. Use of CF has also been linked to poorer quality of life (QoL) in both enterally fed children and their caregivers [7,8,9].

An alternative to CF is blenderised tube feeds (BTF). BTF describes the provision of whole foods including fruits, vegetables, meats, grains, and liquids which have been pureed and administered via a feeding tube [10]. The use of BTF has been controversial and many professional bodies currently do not support BTF due to the potential for increased risk of bacterial contamination, nutritional inadequacy, tube blockages, and limited research on clinical outcomes on BTF [11]. Despite limited support from health practitioners, BTF has become a popular alternative to CF in recent years [12,13]. More parents are requesting to incorporate “real food” into their children’s feeding regimen. This is due to a cultural shift towards consuming natural foods, concerns regarding the dietary quality of formula, claims that BTF reduces adverse symptoms of tube feeding, and to “normalise feeding” [14,15].

In recent years some small studies have emerged which exemplify benefits of BTF, including decreased gastrointestinal (GI) symptoms, improved feed tolerability, increased gut microbial diversity, and improved QoL [5,7,16,17,18,19]. However, clinical and nutritional information on BTF is limited and further investigation is needed to enable clinicians to make recommendations on BTF use. This prospective study primarily sought to clinically characterise and assess patient outcomes of gastrostomy-fed children receiving BTF compared to children receiving CF. This study also investigated the secondary outcomes of intestinal inflammation and intestinal microbiome composition.

## 2. Materials and Methods

### 2.1. Patient Recruitment

Children aged 2–18 years who receive EN via gastrostomy feeds at Sydney Children’s Hospital, a tertiary paediatric hospital in Sydney, Australia, were eligible for this study. Gastrostomy-fed children receiving BTF were identified by the dietitians involved in their care. Children receiving CF were age-, sex-, and disease-matched 1:1 with children receiving BTF using the hospital gastrostomy database. Identified BTF and matched CF children were then invited to join the study. Children were included in the BTF group if ≥25% of their caloric intake was provided by BTF. Participants were classified in the CF group if they were solely formula-fed. Inclusion criteria for the study required participants to be established on gastrostomy feeds for >6 months. Children receiving supplemental parenteral nutrition in addition to gastrostomy tube feeding remained eligible for the study. Children fed via jejunal or nasogastric tube, children who received ≥10% of their caloric intake orally, and children with multiple food allergies or requiring specialised diets due to underlying disorders were excluded from the study. Examples of specialised diets which were excluded from the study included ketogenic diets, diabetic diets, diets specialised for renal disease, and fluid restrictions for cardiac disease.

Participants were recruited and assessed at a single time point, and all data were collected between May 2020 and August 2020 for both groups. Informed consent was obtained from caregivers of all study participants, with individual subject consent as appropriate. Demographic information such as age, sex, underlying comorbidities, time since gastrostomy insertion, medication, allergies, and recent procedures were recorded based on information gathered from Electronic Medical Records (EMR). Outcome measures for GI symptoms and growth/nutrition were compared between the BTF and CF groups. Post-hoc analysis was conducted based on percentage of caloric intake from BTF. Participants in the BTF group were categorised into the 25–50%, 50%, or 100% BTF subgroups based on information from their treating dietitian. Ethical approval for this study was granted by the Sydney Children’s Hospital Ethics Committee (reference no. 2020/ETH00308 and LNR/2020/STE00501).

### 2.2. Outcomes

To assess GI symptoms, participant’s caregivers completed the validated questionnaire, the Pediatric Quality of Life Inventory Gastrointestinal Symptoms Scale (GI-PedsQL). Each item of the GI-PedsQL uses a 5-point Likert scale from 0 (Never) to 4 (Almost always). The raw scores were reverse-scored and linearly transformed to a 0 to 100 scale (0 = 100, 1 = 75, 2 = 50, 3 = 25, 4 = 0). The mean score for each subscale and the total scale were calculated by summing the scores of all items in the subscales and the total scale divided by the number of items answered [20]. Symptom scores were not calculated if more than 50% of questions in a subsection were unanswered. To assess parental satisfaction with the child’s feeding regimen, participants’ caregivers completed a Likert scale for satisfaction with enteral regimen (1–5, with 5 being most satisfied). Parental experiences with tube feeding as shared upon recruitment interviews were recorded.

To assess growth, weight, height, and body mass index (BMI) z-scores were recorded from participants’ most recent hospital visit within the last six months (performed November 2019 through May 2020) as available in medical records. Nutritional status of participants was assessed by obtaining participants’ most recent nutritional blood markers completed within the last six months (performed November 2019 through May 2020) as available in hospital medical records.

Dietary intake data were obtained from a detailed 24-h food recall form completed by the primary caregiver. Nutritional analysis was conducted using Foodworks10, a professional nutrition analysis software (FoodWorks 10 Professional, v10.0. 2019 Xyris Pty Ltd., Brisbane, Australia). Vitamin and mineral intake were compared with age- and sex-appropriate dietary reference intake; recommended dietary intake (RDI) for B_12_, folate, calcium, vitamin D, vitamin C, vitamin A, magnesium, iron, selenium, and zinc, and adequate intake (AI) for potassium, vitamin E, and fibre as based on nutrient reference values for Australia and New Zealand [21]. 

Stool sample collection kits were sent to all study participants. Participants could choose to post the collected sample within 24 h of collection to the laboratory via express mail or directly drop the sample to the laboratory. The timing of stool sample collection was at the patient and caregiver’s convenience and were not collected at the same time as the completion of the GI-PedsQL and obtainment of the anthropometric and nutritional data. Caregivers were advised to avoid collecting stool samples whilst their child was unwell or if they were completing a course of antibiotics to reduce the incidence of non-diet-related factors which may influence stool analysis results.

Once received into the laboratory, stool samples were aliquoted for microbiome and faecal calprotectin (FC) testing and stored at −80 °C until analysis. FC levels were assessed using the PhiCal ELISA using the manufacturer’s protocol (CALPRO, Svar Life Science group, Lysaker, Norway). Microbiome analysis was conducted as follows: Total DNA was extracted from the faecal samples using QIAamp PowerFecal DNA Kits (Qiagen Australia, Vic, Australia) as per manufacturer’s protocol. 16S rRNA sequencing was undertaken at the Ramaciotti Centre for Genomics (UNSW, Sydney, Australia) on the Illumina MiSeq platform (Ilumina Inc., San Diego, California, USA) using V3-V4 hypervariable regions of bacterial 16S rRNA gene (forward primer (341F): CCTACGGGNGGCWGCA and reverse primer (805R): GACTACHVGGGTATCTAAT CC). Forward and reverse sequences were trimmed, denoised, filtered, and grouped into amplicon sequence variants in RStudio version 1.4.1717 (RStudio Inc., Boston, MA, USA). Low abundance and chimeric sequences were removed, and microbial taxonomy was assigned to species classification where possible using the Silva NR99 v138 16S rRNA dataset with Shannon diversity calculated and raw abundance converted to relative abundance for analysis.

### 2.3. Statistical Analysis

Descriptive statistics were used to summarise the data. Categorical variables between the groups were analysed using Fisher’s exact test. Continuous variables between the groups were analysed using *t*-test for parametric variables and the Mann–Whitney U test for non-parametric variables. BTF subgroup analyses were conducted using an ordinary one-way analysis of variance (ANOVA) with Tukey’s multiple comparisons post-hoc test. A *p* value < 0.05 was set as the threshold for statistical significance. For a two-tailed *t*-test with alpha = 0.05, the achieved power for the sample size of the study was 99.8%. Statistical analyses were conducted using GraphPad Prism 8.0 (GraphPad Software, Inc., San Diego, CA, USA).

## 3. Results

### 3.1. Demographics

A total of 21 participants receiving BTF and 20 receiving CF were recruited into the study. The groups did not differ in age, sex, underlying comorbidity, or time since gastrostomy insertion (Table 1). Three patients were also receiving parenteral nutrition (PN) (*n* = 2 BTF, *n* = 1 CF). Of the children receiving parenteral nutrition, two children had a PN:EN ratio of 2:1 and one child had a PN:EN ratio of 1:1. All children in the BTF group had been consistently using BTF for ≥6 months. Within the BTF group, percentage of caloric intake from BTF varied (Table 1). Use of proton pump inhibitors (PPIs) for gastroesophageal reflux disease (GORD), stool softeners for constipation, and anti-diarrhoeals was not different between the groups. Use of pro-kinetic drugs was higher in the CF group (Table 1).

### 3.2. GI Peds-QL Questionnaire Scores

A total of 38 participants (*n* = 20 BTF and *n* = 18 CF) completed the GI-PedsQL. Children on BTF reported better GI symptom scores (73.057 vs. 47.97, *p* = <0.0001) and total scores (74.65 vs. 50.125, *p* = 0.004) on the GI-PedsQL, indicating significantly fewer GI symptoms. All except two of the GI-PedsQL sub scores were significantly greater in the BTF group, indicating fewer complaints of stomach upset, heartburn/reflux, nausea/vomiting, and diarrhoea and constipation (Table 2). Trouble swallowing and communication issues did not differ between the groups, likely due to a similar number of children in both groups having neuromuscular impairment. Sub-section scores for worries about symptoms and communication were not completed for 20 participants (*n* = 7 BTF, *n* = 13 CF) as many participants had developmental delay or were non-verbal. 

A one-way ANOVA showed that the effect of percentage caloric intake from BTF on GI symptoms was significant, F (3,31) = 7.617, *p* = 0.0006. Post-hoc multiple comparisons testing showed that GI symptom scores were significantly better in the 50% and 100% BTF groups compared to the CF group and comparable between the BTF subgroups (Table 3), indicating that partial use of BTF is sufficient to significantly improve GI symptoms.

### 3.3. Anthropometrics

Anthropometric data for the BTF and CF groups are shown in Table 4. Median time since measurement of height and weight was 3.7 months, ranging from 0.5 to 5.6 months. There was no significant difference in height between the BTF and CF groups. The CF group had significantly higher weight and BMI z-scores compared to the BTF group (Table 4). Incidence of malnutrition, defined as weight-for-height z-scores below −2, was 20% in the BTF group and 4.8% in the CF group, although this was not significant. One child in the CF group and four the BTF group were malnourished. The four malnourished children in the BTF group all had cerebral palsy and the child in the CF group had a genetic syndrome.

A one-way ANOVA comparing percentage of caloric intake from BTF and BMI showed that the effect of percentage BTF intake on BMI was significant, F (3,31) = 9.6, *p* = <0.001. Post hoc comparisons using Tukey’s test indicated that BMI z-scores were significantly higher in the 25% and 50% BTF groups, compared to children on 100% BTF (Figure 1). BMI z-scores were comparable between children on 25% and 50% BTF and children on CF, indicating that partial BTF is not associated with poorer growth (Table 5).

Post hoc analysis of the BTF group based on time since commencement of BTF found that BMI z-scores were poorer with increased duration on BTF (Figure 2A). Results were significant between children on BTF for ≥2 years and children on CF (−1.357 vs. 0.5300, *p* = 0.021). Post hoc analysis dividing the BTF group by age indicated that BMI z-scores decreased with increased age of child on BTF (Figure 2B). Children >10 years of age on BTF had significantly poorer BMI z-scores than children on CF (−1.798 vs. 0.530, *p* = 0.035), indicating that BTF may not support growth in older children.

### 3.4. Nutritional Biochemistry

Nutritional biochemistry did not vary significantly between both groups (Table 6). All study participants fell within the normal range for all blood marker components. The blood marker components assessed included albumin, corrected calcium, ferritin, magnesium, phosphate, haemoglobin, vitamin D, folate, vitamin B_12_, C-reactive protein (CRP), zinc, haematocrit, and erythrocyte sedimentation rate (ESR) (Table 6). However, complete nutritional blood marker panels were not available for all participants. Median time since measurement of blood results was 4.2 months, ranging from 0.3 to 5.9 months. Several children received supplemental multivitamins (*n* = 4 BTF, *n* = 3 CF), vitamin D (*n* = 5 BTF, *n* = 4 CF), and iron (*n* = 6 BTF, *n* = 6 CF), which likely impacted blood marker results.

### 3.5. Nutritional Analysis of Feeds

Full diet histories were available for 31 participants (*n* = 12 BTF, *n* = 19 CF). The BTF group received significantly greater energy per kilogram of body weight from feeds (Table 7). In all participants, caloric intake was sufficient to meet their estimated energy requirements (EER), as calculated by the treating dietician. Macronutrient distribution varied significantly between the two groups. Both groups had a similar fat intake, but the BTF group had higher protein and lower carbohydrate intake than the CF group (Table 7). Dietary fibre intake was significantly greater in the BTF diets (23.88 ± 10.25 g vs. 3.97 ± 6.08 g, *p* = <0.001). Levels of certain micronutrients such as folate, magnesium, vitamin A, and selenium were significantly higher in the BTF group, however, both groups exceeded RDI (Appendix A). Potassium intake did not meet adequate intake levels in the CF group and was significantly higher in the BTF group.

### 3.6. Stool Sample Analysis

A total of 18 stool samples were collected from study participants: 8 from the BTF group and 10 from the CF group. At the phyla level, the BTF samples appears to have a more consistent microbiome composition for both the *Firmicutes* and *Bacteroidetes phyla*, whereas the CF samples have a more variable composition. Furthermore, the *Firmicutes* to *Bacteroidetes* ratio was overall higher in the CF group although the difference was non-significant. There were no significant differences in the intestinal microbiome composition at the phyla level between the two groups. Alpha diversity assessed by Shannon diversity index was not significantly different between children receiving CF and BTF; however, diversity trended to be higher as the percentage of BTF increased (Figure 3A). Faecal calprotectin (FC) was significantly lower for children receiving BTF (median BTF 33.3 mg/kg vs. median CF 72.3 mg/kg, *p* = 0.043) (Figure 3B). There was a positive correlation between FC and the total GI symptom score (Spearman *r*-0.5539 *p* = 0.0230) (Appendix A).

### 3.7. Parental Satisfaction with Feeding Regimen

Likert scale completion addressing parental satisfaction with feeding regimen was available for all study participants (*n* = 21 BTF, *n* = 19 CF). Results indicated significantly greater satisfaction in caregivers of children on BTF in comparison to caregivers of children on CF (4.905 ± 0.301 vs. 3.789 ± 1.134, *p* = <0.001). Greater scores indicated greater satisfaction with feeding regimen (scale 1–5, 5 being most satisfied). 

## 4. Discussion

### 4.1. Gastrointestinal Symptoms

This study found that children receiving BTF report fewer GI symptoms compared to children receiving CF. Specifically, symptoms of stomach upset, constipation, and nausea/vomiting were significantly less in children on BTF. These results are consistent with other small paediatric studies which have found that children on BTF have less GI symptoms than formula-fed children [5,7,17,18].

Reflux is often the largest complaint for formula-fed children [22]. In addition to superior GI-PedsQL scores for reflux and nausea/vomiting, this study found that medication usage for reflux decreased with BTF. Use of PPIs and pro-motility drugs use were 22% and 31% lower in the BTF group, respectively, reaching statistical significance for pro-motility drugs and complementing findings from Gallagher et al. [5]. Less anti-reflux medication usage and decreased scores for reflux in the BTF group suggests that BTF may decrease reflux. 

The mechanism by which BTF improves GI symptoms is unclear. Theories include that the increased viscosity of BTF compared to CF may decrease the rate of gastric emptying and therefore reduce symptoms of dumping syndrome [18]. Gallagher et al. [5] found that BTF improves gut microbiome diversity and health, which is associated with decreased gut microbial dysbiosis and inflammation [23,24]. Gut microbiome analysis in the current study suggested more a consistent microbiome composition with respect to the *Firmicutes* to *Bacteroidetes*
*phyla* distribution, and lower gastrointestinal inflammation with BTF, with faecal calprotectin levels, a marker of gastrointestinal inflammation being significantly lower in the BTF group compared to the CF group. Gut microbiome diversity was not significantly different between the CF and BTF groups although there was a trend towards increasing diversity with an increasing percentage of BTF in the diet. This may be due to type II error resulting from the small sample set (*n* = 4)) of participants receiving 100% BTF who were included in the microbiome analysis. Nevertheless, these results are consistent with BTF improving GI symptoms likely secondary to improved gut microbiome health although further research is needed in this area. Research comparing pH-impedance to evaluate reflux and electrogastrography and gastric emptying studies to investigate gastric function between patients on BTF and CFs may also help illustrate how BTF improves GI symptoms.

No research has previously been conducted to determine the caloric intake needed from BTF to reduce GI symptoms. This is the first study which shows that 50% caloric intake from BTF significantly improves GI symptoms compared to solely formula-fed children. Whilst GI symptom scores were better in the 25% BTF group compared to the CF only group, results were not significant for all symptoms given the small sample size. Symptoms were significantly better in the heartburn/reflux and nausea/vomiting domains. Results were significantly better in the 50% and 100% BTF groups when compared to the CF group, and similar between the 50% and 100% BTF groups, indicating that partial BTF use may be sufficient to improve adverse GI symptoms associated with tube feeding. Hence, this study provides a rationale to conduct further studies to determine the amount of BTF required to improve GI symptoms associated with tube feeding.

### 4.2. Growth

The literature is currently divided on the impact of BTF on growth, however, a concern associated with BTF use is impaired growth due to the variable caloric content of homemade feeds [25,26]. In this study, the BTF and CF cohorts had similar z-scores for height; however, the BTF group had lower z-scores for weight and BMI as well as higher levels of malnourishment. This suggests that BTF alone may be inadequate to support growth in children and highlights the importance of frequent dietitian monitoring for children on BTF.

Subgroup analyses revealed that children on BTF for greater than two years had significantly lower BMIs compared to formula-fed children, suggesting that long-term BTF may have a detrimental effect on growth. Additionally, children older than 10 years had poorer BMIs than children on CF, indicating that BTF may not support growth in older children who have higher caloric requirements due to pubertal growth spurts. However, older children were likely to have been on BTF for longer and thus it is unclear if both factors are independently associated with poorer BMI. Further, BMI does not consider body composition and for some clinical conditions and can be an unreliable measure of growth status. Orel et al. [27] found that children on CF had a greater increase in BMI than children on BTF; however, children on CF had a greater increase in fat mass as opposed to lean body mass in comparison to children on BTF.

Thus, consideration needs to be made to use other anthropometric measures to assess growth including tricep skinfold thickness, mid-upper arm circumferences, and bioelectric impedance analysis. Regardless, these findings highlight the importance of frequent dietetic review for children on BTF for extended periods to ensure adequate caloric intake to support growth. Additionally, future studies should investigate whether growth outcomes are improved for patients receiving BTF based on the regularity of their dietitian reviews, as infrequent reviews may account for the inadequate weight gain on BTF, as the literature suggests that families may often not reach out to dietitians for assistance for BTF due to previous negative experiences [28].

This was the first study to evaluate the association between the percentage of caloric intake from BTF and growth. Subgroup analysis demonstrated that BMI z-scores did not differ significantly for children receiving 25% and 50% BTF in comparison to children on CF but did so for children on 100% BTF. These results provide a rationale for combined BTF and CF use to reduce tube feeding symptoms without compromising growth. Further research is needed to determine the optimal ratio of BTF to CF.

Given that the daily caloric content of BTF is variable, it has been suggested that BTF compromises growth due to caloric insufficiency [25,29]. However, nutritional analysis in the current study indicated that children on BTF received 1.4-fold greater energy intake than formula-fed children. Notably, Gallagher et al. [5] reported a need for a 1.5-fold increase in calories in children and Tanchoco et al. [30] reported a need for a 1.2-fold caloric increase in adults post-transition from CF to BTF to sustain growth. In this study, increased caloric intake did not appear adequate to maintain weight on BTF; however, dietary intake was only recorded at one point in time, and it is difficult to ascertain if participants had a consistently increased caloric intake. In addition, some of the caloric content of BTF is from dietary fibre which is not readily absorbed.

It is unclear why additional calories are required on BTF, however possible explanations include differences in the thermic effect of feeding between natural and processed foods [5]. Whilst children in this study had poorer growth than their CF counterparts despite a higher caloric, as Gallagher et al. [5] found that children required a 1.5-fold increase in calories whilst on BTF to maintain weight, it is possible that weight loss due to either the thermic effect of feeding or decreased caloric availabilities in natural foods may be counteracted by increasing caloric consumption on BTF. Further studies are needed to quantify the ideal increased caloric intake when transitioning from CF to BTF.

Notably, whilst Gallagher et al. [5] found that a caloric increase was necessary to maintain growth on BTF, the study also found that total body fatness, measured by tricep skinfold measurements increased with BTF use, suggesting that caloric needs may have been overestimated.

### 4.3. Nutrition

Nutritional blood marker results indicated no nutritional deficiencies in either BTF or CF group. Macronutrient distributions of both BTF and CF fell within the recommended range of dietary guidelines for children [31]. Macronutrient ratios varied between the groups, with the BTF group receiving a greater proportion of protein (19% vs. 11%) and a lower proportion of carbohydrates (41% vs. 50%) than the CF group.

Contrary to concerns in the literature that the micronutrient content of BTF is inadequate and highly variable [26,29], micronutrient levels were found to be higher in the BTF group in our study. This is consistent with other recent studies which suggest that BTF diets are an excellent source of nutrition for enterally fed children [32]. Notably, BTF was far superior in fibre content, containing almost 5-fold the amount of fibre in formula. Dietary fibre intake is associated with increased gut microbial diversity and contributes to short-chain fatty acid production in the gut, which regulates immune function and inflammation [33,34]. BTF also has higher dietary fibre content, which creates beneficial changes to gut microbiota composition and bulks stools resulting in decreased constipation [33,34]. Thus, this study demonstrates that complete nutrition with adequate intake of micro- and macronutrients, vitamins, and trace elements is possible on BTF. Further studies on clinically tested BTF recipes including an evaluation of nutrients would be beneficial and may aid in the development of a recipe bank to assist caregivers wishing to use BTF [35].

### 4.4. Stool Microbiota

The relationship between gut microbiome composition and health has garnered much attention in recent years. In this study, children on BTF generally had a healthier gut microbiome than their formula-fed counterparts. Similar to Gallagher et al. [5], this study found that alpha diversity trended to be better in children on BTF. Children on BTF appear to have a healthier, more diverse gut microbiota than CF children as BTF as indicated by the lower *Firmicutes* to *Bacteroidetes* ratio in BTF, which is predominantly due to more *Bacteroidetes* present in BTF. This is likely a result of the consumption of a diverse diet comprised of fresh, whole foods that are high in fibre, and lack of additives that are potentially harmful to gut microbiota [23,36]. A diverse gut microbiota is important in children as it is associated with decreased risk of bowel disease, cardiovascular disease, asthma, allergies, and obesity [37,38].

Reduced levels of *Firmicutes* and *Bacteroidetes* have been associated with decreased bowel inflammation [24]. Thus, children on BTF may experience fewer GIT symptoms due to beneficial changes in the gut microbiome with the use of whole foods which reduce gut inflammation. This is consistent with faecal calprotectin levels, a marker of GIT inflammation, which was significantly lower in the BTF group. Furthermore, there was a correlation between faecal calprotectin and stool microbiome results with GI-PedsQL, suggesting that decreased intestinal inflammation may be the reason for less GIT symptoms and thereby improved QoL in children on BTF. This study highlights that BTF use is beneficial to the gut microbiome.

### 4.5. Caregiver Experiences

In keeping with the literature [5,18,39], parents of children on BTF reported greater satisfaction with their child’s feeding regimen than parents of formula-fed children. Medical reasons parents cited for satisfaction with BTF included that their children had fewer adverse GI symptoms associated with tube feeding, reduced oral aversion and less illness. Whilst oral aversion was not assessed in the current study, oral feeding which was 22% higher in the BTF group trended towards significance and other studies have reported increased oral intake in children post-transition to BTF [5,18]. Tube feeding has been associated with poorer QoL for both children and families of enterally fed children [39,40,41,42]. Psychosocial benefits of BTF use parents reported included that it is a more natural way of feeding, enables them to prepare meals for their child, and allows their child to be included in family mealtimes.

### 4.6. Limitations

Limitations to this study included the relatively small sample size which prevented general assumptions regarding BTF use. However, most results reached statistical significance and the sample size is similar to other studies on BTF. Data collected on GI symptoms from the GI-PedsQL questionnaire were reliant on parental interpretation of their child’s symptoms, as most study participants were non-verbal or developmentally delayed. However, this is equally applicable to both BTF and CF groups and objective measures of gastric function such as medication use to relieve gastrointestinal symptoms also showed better results in the BTF group. Some questionnaires and food-recall forms were not completed to a standard suitable for analysis, and stool samples were not received from all participants impeding sample size and analysis. Another limitation was the fact that the symptom/QoL questionnaires and stool samples were not collected from both the BTF and CF cohorts at baseline and after commencement of the respective feeds. Similarly, data on medication use to treat GI symptoms prior to commencing BTF were not collected. This is something we hope to address in a future randomised cross-over study. Additionally, this study did not compare rates of tube blockages, infection rates, and cost of feeds between the two cohorts.

However, the strengths of the study include a prospectively recruited, medically complex cohort of children established on BTF for ≥6 months who were age- and disease-matched with a group of formula-fed children to limit the effect of confounding factors on findings. The use of a validated questionnaire, the GI-PedsQL, to assess gastrointestinal symptoms in participants was another strength. 

## 5. Conclusions

Given the growing interest in BTF for enterally fed children, it is important to investigate the effect of BTF on symptoms, QoL, growth, and nutrition. This study demonstrates that BTF use is associated with reduced GI symptoms associated with tube feeding. Despite being nutritionally superior to commercial formula in caloric and micronutrient content, careful dietetic monitoring is required to ensure growth goals are met on BTF. Whilst larger, prospective studies are needed, this study provides a basis for investigating if a combination of BTF and CF may help reduce symptoms, whilst supporting growth in medically complex children. 

## Figures and Tables

**Figure 1 nutrients-14-03139-f001:**
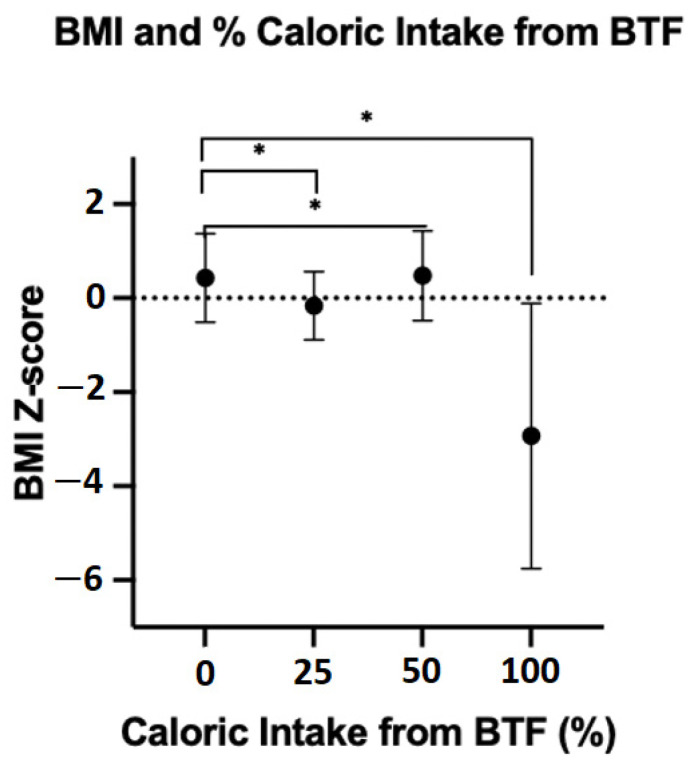
Comparison of BMI z-scores between BTF percentage caloric intake subgroups. BMI, Body Mass Index. CF group is represented as 0% BTF intake. Results are mean BMI z-score. Error bars indicate SD. * Statistically significant (*p* < 0.05). BTF, blenderised tube feeds; CF, commercial formula; SD, standard deviation.

**Figure 2 nutrients-14-03139-f002:**
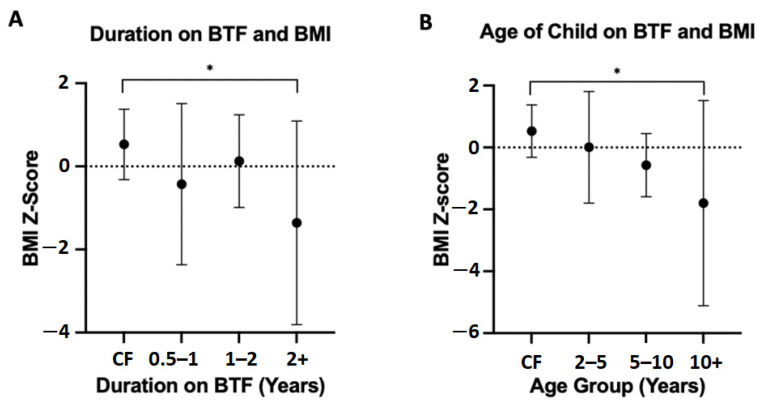
Subgroup analyses of children on BTF and BMI vs. children on CF mean BMI z-score of children on CF and children on BTF divided by years on BTF. Results are from Tukey’s multiple comparisons test. (**A**) Mean BMI z-score of children on CF and children on BTF divided by age of child in years (**B**). BMI, Body Mass Index. * Statistically significant (*p* < 0.05).

**Figure 3 nutrients-14-03139-f003:**
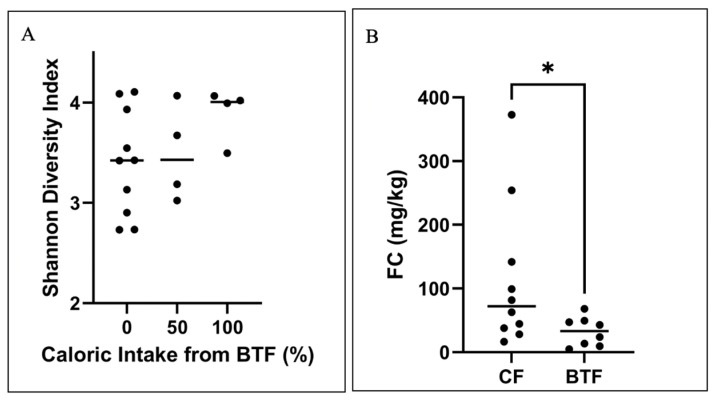
Stool sample analysis results. (**A**) Shannon diversity of faecal samples in children receiving blenderised tube feeding (BTF) or commercial formula (represented as receiving 0% BTF). Line indicates the mean. (**B**) Faecal calprotectin (FC) levels in children receiving blenderised tube feeding (BTF) or commercial formula (CF) feeding * *p* = 0.043 (Mann–Whitney U test). Line indicates median.

**Table 1 nutrients-14-03139-t001:** Demographics of study population.

Characteristics	BTF (*n* = 20)	CF (*n* = 21)	*p* Value
Age (years)	7.42 ± 4.24	6.54 ± 3.05	0.672
Male	13 (65%)	13 (61.9%)	0.990
Time since G-Tube Insertion (years) ^1^	5.194 ± 3.438	4.095 ± 2.370	0.368
**Underlying Comorbidity**			
Neurological Impairment	15 (71.43%)	10 (50%)	0.215
Gastrointestinal Disease ^2^	9 (42.86%)	4 (20%)	0.186
Cystic Fibrosis	0 (0%)	2 (10%)	0.233
OA-TOF ^3^	1 (4.76%)	2 (10%)	0.614
Other	2 (9.52%)	3 (15%)	0.669
**Feeding Method**			
Continuous	0%	6 (30%)	0.009 *
Bolus	13 (61.90%)	7 (35.00)	0.124
Mixed	8 (38.10%)	7 (23.33%)	0.351
Oral	12 (57.14%)	7 (35.00%)	0.215
**Medications**			
PPI ^4^	8 (38.1%)	12 (60%)	0.211
Pro-Kinetics	2 (9.52%)	8 (40%)	0.037 *
Stool Softener/Laxatives	2 (9.52%)	7 (35%)	0.072
Anti-Diarrhoeal	2 (9.52%)	1 (5%)	>0.999
**Average Duration of Time on BTF (years)**			
2.24			
**% Caloric Intake provided by BTF**			
25–50%	4 (20%)		
50%	9 (45%)		
100%	7 (35%)		

^1^ G-Tube, Gastrostomy tube. ^2^ Gastrointestinal disease included inflammatory bowel disease and short-gut syndrome. ^3^ OA-TOF, Oesophageal Atresia-Tracheo-Oesophageal Fistula. ^4^ PPI, Proton Pump Inhibitor. Results are mean ± SD. * Statistically significant (*p* < 0.05). BTF, blenderised tube feeds; CF, commercial formula; SD, standard deviation.

**Table 2 nutrients-14-03139-t002:** Comparison of GI-PedsQL results between children on BTF and CF.

Peds-QL Section	BTF (*n* = 20)	CF (*n* = 18)	*p* Value
Stomach Pain	78.24 ± 16.63	58.10 ± 24.45	0.0051 *
Stomach Upset	83.89 ± 15.84	54.44 ± 29.55	0.0004 *
Food and Drink Limits	47.22± 43.84	24.07 ± 35.66	0.0952 *
Trouble Swallowing	36.11 ± 36.25	29.17 ± 27.63	0.5271
Heartburn/Reflux	77.43 ± 16.93	50.00 ± 29.67	0.0016 *
Nausea/Vomiting	81.60 ± 21.54	36.11 ± 22.11	<0.0001 *
Gas	70.83 ± 18.15	47.69 ± 24.70	0.0027 *
Constipation	79.70 ± 19.57	49.60 ± 27.90	0.0003 *
Blood in Stool	98.61 ± 5.735	70.31 ± 39.49	0.0046 *
Diarrhoea	83.93 ± 20.92	64.18 ± 23.67	0.0282 *
Total GI Symptoms ^1^	73.76 ± 10.41	48.61 ± 17.16	<0.0001 *
Worry about Stool	89.23 ± 89.23 (*n* = 13)	59.07 ± 37.07 (*n* = 9)	0.0200 *
Worry about Abdominal Pain	89.58 ± 19.82 (*n* = 12)	53.13 ± 36.44 (*n* = 8)	0.0095 *
Medications	93.33 ± 12.38 (*n* = 15)	70.54 ± 40.30 (*n* = 7)	0.0545
Communication	42.33 ± 39.95 (*n* = 15)	39.38 ± 35.60 (*n* = 8)	0.8625
Total Score	74.65 ± 8.136 (*n* = 12)	50.125 ± 27.44 (*n* = 7)	0.004 *

^1^ GI, Gastrointestinal. Results are mean ± SD. * Statistically significant (*p* < 0.05). For GI-PedsQL, greater scores indicate less limitation. Peds-QL, Pediatric Quality of Life Inventory; GI-PedsQL, Paediatric Quality of Life Inventory Gastrointestinal Symptoms Scale.

**Table 3 nutrients-14-03139-t003:** Comparison of GI-PedsQL results between % BTF Intake Subgroups and CF Group.

% BTF Groups	Mean Difference	Mean ^1^	Mean ^1^	Standard Error	*p* Value	Confidence Interval	
						Upper Limit	Lower Limit
CF vs. 25	−23.20	51.26	74.46	7.786	0.0553	−41.51	0.7520
CF vs. 50	−23.85	51.26	75.11	5.804	0.0012 *	−40.19	−8.692
CF vs. 100	−20.12	51.26	71.38	6.668	0.0227 *	−38.81	−2.619
25 vs. 50	−0.6488	74.46	75.11	8.370	0.9999	−26.78	18.65
25 vs. 100	3.079	74.46	71.38	8.991	0.9891	−24.74	24.07
50 vs. 100	3.728	75.11	71.38	7.341	0.9566	−16.20	23.65

^1^ Results are from Tukey’s multiple comparisons test. * Statistically significant (*p* < 0.05). For GI-PedsQL, greater scores indicate less limitation.

**Table 4 nutrients-14-03139-t004:** Comparison of anthropometric measurements between BTF and CF groups.

Characteristics	BTF (*n* = 20)	CF (*n* = 21)	*p* Value
Height Z-Score	−1.239 ± 2.419 (*n* = 17)	−1.660 ± 1.047	0.905
Weight Z-Score	−1.722 ± 2.140	−0.506 ± 1.38	0.0108 *
BMI Z-Score	−0.597 ± 2.099 (*n* = 17)	0.428 ± 0.8455	0.045 *
% Malnourished	4 (20%)	1 (4.76%)	0.183

Results are mean ± SD. BMI, body mass index. * Statistically significant (*p* < 0.05).

**Table 5 nutrients-14-03139-t005:** Comparison of BMI z-score by % BTF intake groups.

% BTF Groups	Mean Difference	Mean ^1^	Mean ^1^	Standard Error	*p* Value	Confidence Interval	
						Upper Limit	Lower Limit
CF vs. 25	0.5905	0.4280	−0.1625	0.6796	0.8207	−0.6856	2.610
CF vs. 50	−0.04771	0.4280	0.4757	0.5449	0.9998	−1.395	1.504
CF vs. 100	3.361	0.4280	−2.933	0.6796	<0.0001 *	1.659	5.266
25 vs. 50	−0.6382	−0.1625	0.4757	0.7777	0.8442	−2.827	1.012
25 vs. 100	2.770	−0.1625	−2.933	0.8773	0.0176 *	0.3016	4.699
50 vs. 100	3.408	0.4757	−2.933	0.7777	0.0007 *	1.354	5.463

^1^ Results are from Tukey’s multiple comparisons test. * Statistically significant (*p* < 0.05).

**Table 6 nutrients-14-03139-t006:** Nutritional biochemistry in children on BTF vs. CF.

Component	BTF	CF	*p* Value
Albumin	38.12 ± 4.512 (*n* = 17)	41.86 ± 22.81 (*n* = 14)	0.7908
Corrected Calcium	2.398 ± 0.111 (*n* = 16)	2.356 ± 0.0829 (*n* = 13)	0.2691
Ferritin	85.00 ± 157.3 (*n* = 15)	51.30 ± 41.66 (*n* = 12)	0.5451
Magnesium	0.879 ± 0.124 (*n* = 16)	0.8292 ± 0.099 (*n* = 13)	0.6257
Phosphate	1.498 ± 0.204 (*n* = 14)	1.388 ± 0.253 (*n* = 12)	0.2301
Haemoglobin	126.1 ± 12.15 (*n* = 15)	121.3 ± 19.27 (*n* = 15)	0.4151
Vitamin D	88.91 ± 25.33 (*n* = 13)	85.45 ± 25.54 (*n* = 13)	0.7534
Vitamin B_12_	651.8 ± 333.0 (*n* = 6)	559.2 ± 371.1 (*n* = 7)	0.6476

Results are mean ± SD.

**Table 7 nutrients-14-03139-t007:** Comparison of nutritional composition of BTF and CF diets.

Nutrients	BTF (*n* = 12)	CF (*n* = 19)	*p* Value
kJ/kg	325.4 ± 134.1	242.5 ± 261.1	0.016 *
Total KJ	6436 ± 1711	5227 ± 1447	0.043 *
% kJ from Carbohydrates	41.25 ± 7.208	49.90 ± 6.716	0.002 *
% kJ from Protein	19.03 ± 5.004	10.75 ± 2.332	<0.001 *
% kJ from Fat	37.53 ± 4.775	38.99 ± 5.447	0.818
Carbohydrate Total (g)	157.4 ± 49.66	144.5 ± 59.20	0.536
Protein (g)	72.34 ± 24.51	32.62 ± 13.50	<0.001 *
Fat (g)	64.79 ± 20.00	53.23 ± 18.54	0.112
Fibre (%RDI ^1^)	119.4 ± 51.25	19.86 ± 30.38	<0.001 *

^1^ RDI, Recommended Dietary Intake. Results are mean ± SD. * Statistically significant (*p* < 0.05).

## Data Availability

Not applicable.

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
