# Peer review of "Blenderised Tube Feeds vs. Commercial Formula: Which Is Better for Gastrostomy-Fed Children?"

_nutrients, 2022, doi:10.3390/nu14153139_

Round 1

Reviewer 1 Report

The Author proposed a relevant issue regarding tube-feeded children. They compare patients on blenderized formula feed to patients on commercial formula, sharing a relevant set of data.

Their main findings are:

- Caregivers of children on BTF reported greater GI-PedsQL scores indicating significantly less GI symptoms: this is a subjective parent-reported score and I can imagine that parents willing to feed their child with natural food may inadvertently report their child's symptoms with less authenticity to support their dietary choice

Faecal calprotectin levels were significantly lower for children receiving BTF compared to children on CF and the BTF group had healthier, more diverse gut microbiota. Calprotectin is a non specific marker of intestinal inflammation. Despite suggestive of a more healthy microbiota, I have some difficulties in accepting this evidence on the basis of a single evaluation on patients with different condition requiring tube feed and a very variable length of tube feed. The authors should have performed a initial analysis (t0) followed by a t1 re-evaluatio. Moreover, too many factors other than diet can influence microbiota composition (age, antibiotics, enviromental,...) and we have not data on this aspect regarding the two population. Figure 4 is not very helpful in understanding microbiota diversity for readers not very familial with this issue. 

- Subgroup analysis found that 25% of caloric intake from BTF was sufficient to improve GI symptoms. Interesting, but we have nor informations on the previous GI symptoms reported by the parents at the beginning of tube feed, neither of drug requirement. How many patients on both groups were on PPI, how many on prokinetics before starting tube feed? How many patients had reflux? This lacking informations are upmost important, considered the variability of the two populations, composed of a majority of neurologically impaired children in the BTF group and with more cystic fibrosis patients or esophageal atresia in the CF group. I suggest the authors the make their comparison on a more homogeneous group of patients (such as NIC)   

Author Response

Thank you very much for reviewing our paper and for your helpful feedback. 

Comment 1: Caregivers of children on BTF reported greater GI-PedsQL scores indicating significantly less GI symptoms: this is a subjective parent-reported score and I can imagine that parents willing to feed their child with natural food may inadvertently report their child's symptoms with less authenticity to support their dietary choice 

Whilst the GI-PedsQL questionnaire is a subjective-parent reported score, it has been validated in previous studies as an accurate tool to assess GI symptoms in children with chronic diseases. However, weacknowledge this point and have addressed this limitation in our discussion. Additionally, we looked at more quantitative markers of GI function/health such as medication use and stool analysis. The statistically significant, positive correlation between faecal calprotectin levels and GI-PedsQL results in our study supports the findings of better GIT symptoms and health on BTF diets.

Comment 2: Faecal calprotectin levels were significantly lower for children receiving BTF compared to children on CF and the BTF group had healthier, more diverse gut microbiota. Calprotectin is a non specific marker of intestinal inflammation. 

Faecal calprotectin is a well validated marker of intestinal inflammation.  Faecal calprotectin levels have been used as a marker of intestinal inflammation in several studies in the literature including the reference provided in brackets (Ricciuto A, Griffiths AM. Clinical value of faecal calprotectin. Crit Rev Clin Lab Sci. 2019 Aug;56(5):307-320. doi: 10.1080/10408363.2019.1619159. Epub 2019 Jun 6. PMID: 31088326.) Thus, we believe it is a reasonable tool to use as a marker of intestinal inflammation in our study.

Comment 3: Despite suggestive of a more healthy microbiota, I have some difficulties in accepting this evidence on the basis of a single evaluation on patients with different condition requiring tube feed and a very variable length of tube feed. The authors should have performed a initial analysis (t0) followed by a t1 re-evaluation.

We acknowledge that Your comment that our evidence was based on a single evaluation and that we should have “performed an initial analysis (t0) followed by a t1 re-evaluation”,  is a very valid point which we have now discussed as a limitation of our study. However, we conducted a cross-sectional study and hence we were unable to do a bi-phased analysis. We believe that larger, longitudinal studies in which children are transitioned onto BTF and then followed-up at regular intervals are needed and have discussed the need for such studies in our discussion.

Comment 4:  Moreover, too many factors other than diet can influence microbiota composition (age, antibiotics, enviromental,...) and we have not data on this aspect regarding the two population. Figure 4 is not very helpful in understanding microbiota diversity for readers not very familial with this issue. 

It is a very important point that you raise that several factors affect gut microbiome composition independent of diet. We attempted to control for these factors by age, sex and disease matching the BTF and CF groups. The average length of time on gastrostomy feeds was comparable between the two groups. Additionally, we advised parents to avoid collecting stool samples when the child was unwell or if they were completing a course of anti-biotics to further reduce the incidence of non-diet related factors which may influence stool analysis results. Thank you for notifying us that Figure 4 (regarding intestinal microbial diversity) is difficult to interpret for readers unfamiliar with this issue. We have now removed this figure and the results it displayed are already stated in the text.

Comment 5: Subgroup analysis found that 25% of caloric intake from BTF was sufficient to improve GI symptoms. Interesting, but we have nor informations on the previous GI symptoms reported by the parents at the beginning of tube feed, neither of drug requirement. How many patients on both groups were on PPI, how many on prokinetics before starting tube feed? How many patients had reflux?

This is a very important comment and unfortunately given that our study is a cross-sectional study we do not have data on the GI symptoms of children in the BTF group prior to starting BTF. We have now included this as a limitation of the study in our discussion. Notably, anecdotally the majority of parents of children in the BTF group stated that their child’s GI symptoms significantly reduced after transitioning to BTF and this was a leading reason for parents choosing to change to BTF from formula. Similarly, we do not have data on the use of PPIs, prokinetic drugs and apperients in the BTF group, prior to the commencement of BTF. However, this cross-sectional study found that use of these agents was lower in the BTF group in comparison to the CF group. Additionally, parents felt that use of the above GI medications decreased on a BTF diet. As discussed previously, ideally future studies will be longitudinal and assess symptoms and medications at T0 and T1, prior to and after commencing BTF or CF to validate our findings and this has now been mentioned in the study limitations.

Comment 6: This lacking informations are upmost important, considered the variability of the two populations, composed of a majority of neurologically impaired children in the BTF group and with more cystic fibrosis patients or esophageal atresia in the CF group. I suggest the authors the make their comparison on a more homogeneous group of patients (such as NIC).

Regarding your comment on the variability of the BTF and CF groups medical conditions, the two group were well matched comorbidity wise with comparable levels of GIT, neurological and other conditions. Although there were more children with neurological impairment in the BTF group and with cystic fibrosis and OA-TOF in the CF group, the difference in the numbers was small and not statistically significant when comparing both groups as shown in Table 1.

Once again, we thank you for your valuable comments and insights and hope that we have answered/addressed your questions and concerns.

Reviewer 2 Report

In this original article, the authors presented results of a prospective study comparing blenderised Tube Feeds (BTF) and commercial formula (CF) for enterally fed children. A subject of this article is valuable for clinicians, because BTF can be considered as alternative for enteral feeding.

The limitation of this study is a small sample of patients. The authors should report a power of statistical tests used in the presented cohort. In addition, tests used in all presented statistical analyses (in tables and figures) should be indicated.

Duration of BTF and CF should be clearly presented in section „material and methods”. In addition, it is not clear when the presented data were collected. All parameters should be collected in the same time after identical duration of feeding in both groups.

Author Response

Thank you very much for reviewing our paper and for your helpful feedback.

Comment 1: The limitation of this study is a small sample of patients. The authors should report a power of statistical tests used in the presented cohort. In addition, tests used in all presented statistical analyses (in tables and figures) should be indicated.

Thank you very much for reviewing our paper and for your helpful feedback. We agree that the sample size of our patients is a limitation of this study. However, it is still one of the larger studies on BTF in comparison to the current literature on the topic and the majority of our results were statistically significant. Based on your recommendation we have the included power calculations for the study. You suggested that all tests used in the presented statistical analyses should be indicated in the relevant tables and figures. As discussed in our methodology, categorical variables between the groups were analysed using Fisher’s exact test.  Continuous variables between the groups were analysed using T-test for parametric variables and the Mann-Whitney U test for non-parametric variables. Based on your feedback we have indicated the tests used for the presented statistical analysis in the relevant tables and figures, where basic statistical tests were not used.

Comment 2: Duration of BTF and CF should be clearly presented in section „material and methods”.

You indicated that the duration of time on BTF and CF should be clearly presented. The duration of time on CF is the same as the time since gastrostomy insertion as indicated in the demographic’s tables. As per your suggestion, we have now included the duration of time on BTF in this table as well.

Comment 3: In addition, it is not clear when the presented data were collected. All parameters should be collected in the same time after identical duration of feeding in both groups.

Participants were recruited and assessed at a single time point, and all data was collected between May to August 2020 for both groups. We have edited the paper to include this information in a clearer manner. We acknowledge your comment regarding the fact that ideally all parameters should be collected at the same time after an identical duration of feeding in both groups. Whilst all data was collected during the same time period for both groups, all children in the BTF group were initially on CF first and were transitioned to varying degrees of BTF, at varying points in time due to GI symptoms. As our study was a cross-sectional study comparing the data between the groups at a single point in time, we cannot feasibly control for the duration on each feed type given that almost all children will be started on CF first. However, the average duration of enteral feeds using a gastrostomy was comparable between the groups. We acknowledge a need for larger-longitudinal studies in the future in order to address this issue.

We hope that the changes made are satisfactory and once again thank you for your helpful comments.

Reviewer 3 Report

This is a well written manuscript and will be of interest to the readers.

Minor concerns:

1. The key issue for this MS is the to compare gastrointestinal (GI) symptoms, GI inflammation and stool microbiome composition between children receiving BTF or CF. Additional information can present as Supplementary data. For example, you can put Table 7 as Supplementary table, and Figure 3 as Supplementary figure.

2. So many tables and figures in this manuscript, which will weaken the theme. You can put Figure 5 and Figure 6 together, and put Figure 7 as Supplementary figure. 

3. Add unit of all the measurements in Table 1 (such as Time since G-Tube Insertion should be ‘’Time since G-Tube Insertion, days), and Table 6 ( By the way, no abbreviation for the Hermoglobin in the table, so the Hb in the footnote is not necessary. ).

Author Response

Thank you very much for reviewing our paper and for your helpful feedback. 

Comment 1: The key issue for this MS is the to compare gastrointestinal (GI) symptoms, GI inflammation and stool microbiome composition between children receiving BTF or CF. Additional information can present as Supplementary data. For example, you can put Table 7 as Supplementary table, and Figure 3 as Supplementary figure.

Based on your suggestion we have moved figures and tables displaying results for secondary aims to the supplementary data section. This includes Figure 3 (on micronutrient levels) and figure 7 (on correlation between GI-PedsQL and faecal calprotectin levels) as recommended by you.

Comment 2: So many tables and figures in this manuscript, which will weaken the theme. You can put Figure 5 and Figure 6 together, and put Figure 7 as Supplementary figure. 

We have combined Figure 5 and Figure 6 based on your suggestion. As addressed in the prior comment, we have also moved Figure 7 to the supplementary data section.

Comment 3:  Add unit of all the measurements in Table 1 (such as ‘Time since G-Tube Insertion’ should be ‘’Time since G-Tube Insertion, days’), and Table 6 (By the way, no abbreviation for the Hermoglobin’ in the table, so the ‘Hb’ in the footnote is not necessary. ).

We have rectified the issues noted in Tables 1 and 6 by ensuring that units of all measurements are stated and that unnecessary abbreviations have been removed.

We note that you have indicated that editing of English language is required and have closely reviewed the paper and made appropriate English language corrections. We hope that the changes made are satisfactory and once again thank you for your helpful comments.

Round 2

Reviewer 1 Report

I appreciate that the Authors have recognized the importance of some of my observations and that they have somehow incorporated them in the discussion.

Your reply to my comment #4 on possible factors other than diet that could have influenced the microbiota analysis ("Additionally, we advised parents to avoid collecting stool samples when the child was unwell or if they were completing a course of anti-biotics to further reduce the incidence of non-diet related factors which may influence stool analysis results") should be incorporated in the methods section (page 3, line 120 and following).

Your references list lacks of some important recent papers that could be useful for you to take into account for your discussion. The following could be added and discussed at your convenience:

Behnam Bahramian, Mahboobe Sarabi-Jamab, Saeedeh Talebi, Seyed Mohammad Ali Razavi, Mitra Rezaie

Designing blenderized tube feeding diets for children and investigating their physicochemical and microbial properties and Dietary Inflammatory Index

Nutr Clin Pract . 2022 Jul 12. doi: 10.1002/ncp.10893. Online ahead of print.

Brekke G, Raun AMT, Sørensen SB, Kok K, Sørensen JL, Born AP, Mølgaard C, Hoei-Hansen CE.    Nutrition and preparation of blenderized tube feeding in children and adolescents with neurological impairment: A scoping review.

Nutr Clin Pract. 2022 Aug;37(4):783-796. doi: 10.1002/ncp.10853. Epub 2022 Apr 11.

Soscia J, Adams S, Cohen E, Moore C, Friedman JN, Gallagher K, Marcon M, Nicholas D, Weiser N, Orkin J.

The parental experience and perceptions of blenderized tube feeding for children with medical complexity.

Paediatr Child Health. 2021 Jun 28;26(8):462-469. doi: 10.1093/pch/pxab034. eCollection 2021 Dec.

Chandrasekar N, Dehlsen K, Leach ST, Krishnan U.

Exploring Clinical Outcomes and Feasibility of Blended Tube Feeds in Children.

JPEN J Parenter Enteral Nutr. 2021 May;45(4):685-698. doi: 10.1002/jpen.2062. Epub 2021 Feb 17

Author Response

Thank you very much for the further review of our manuscript. We value your feedback and have made the changes which you have recommended.

  1. Your reply to my comment #4 on possible factors other than diet that could have influenced the microbiota analysis ("Additionally, we advised parents to avoid collecting stool samples when the child was unwell or if they were completing a course of anti-biotics to further reduce the incidence of non-diet related factors which may influence stool analysis results") should be incorporated in the methods section (page 3, line 120 and following).

We have now incorporated this comment regarding the stool sample collection into the methods section in the location which you have suggested. 

2. Your references list lacks of some important recent papers that could be useful for you to take into account for your discussion. The following could be added and discussed at your convenience:

Thank you for this very helpful comment. We acknowledge the importance of discussing the more recent literature which has become available on BTF in this manuscript. We have now incorporated key information from the papers which you have suggested in our discussion.

Once again, we thank you for your very helpful input and hope that you find the changes which we have made to the manuscript satisfactory.

Reviewer 2 Report

The manuscript can be accepted in the present form.

Author Response

Thank you very much for reviewing our manuscript. We have appreciated all of your input and are glad that you find it suitable for publication.